# Overexpression of Heat Shock Protein 70 Improves Cardiac Remodeling and Survival in Protein Phosphatase 2A-Expressing Transgenic Mice with Chronic Heart Failure

**DOI:** 10.3390/cells10113180

**Published:** 2021-11-15

**Authors:** Somy Yoon, Ulrich Gergs, Julie R. McMullen, Gwang Hyeon Eom

**Affiliations:** 1Department of Pharmacology, Chonnam National University Medical School, Hwasun 58128, Korea; 2Institute of Pharmacology and Toxicology, Faculty of Medicine, Martin Luther University Halle-Wittenberg, 06097 Halle, Germany; ulrich.gergs@medizin.uni-halle.de; 3Baker Heart and Diabetes Institute, Melbourne, VIC 3004, Australia; Julie.mcmullen@baker.edu.au

**Keywords:** post-translational modification, phosphorylation, dilated cardiomyopathy, heart failure, HSP70

## Abstract

Heat shock protein (HSP) 70 is a molecular chaperone that regulates protein structure in response to thermal stress. In addition, HSP70 is involved in post-translational modification and is related to the severity of some diseases. Here, we tested the functional relevance of long-lasting HSP70 expression in a model of nonischemic heart failure using protein phosphatase 2 catalytic subunit A (PP2CA)-expressing transgenic mice. These transgenic mice, with cardiac-specific overexpression of PP2CA, abruptly died after 12 weeks of postnatal life. Serial echocardiograms to assess cardiac function revealed that the ejection fraction (EF) was gradually decreased in transgenic PP2CA (TgPP2CA) mice. In addition, PP2CA expression exacerbated systolic dysfunction and LV dilatation, with free wall thinning, which are indicators of fatal dilated cardiomyopathy. Interestingly, simultaneous expression of HSP70 in double transgenic mice (dTg) significantly improved the dilated cardiomyopathy phenotype of TgPP2CA mice. We observed better survival, preserved EF, reduced chamber enlargement, and suppression of free wall thinning. In the proposed molecular mechanism, HSP70 preferentially regulates the phosphorylation of AKT. Phosphorylation of AKT was significantly reduced in TgPP2CA mice but was not significantly lower in dTg mice. Signal crosstalk between AKT and its substrates, in association with HSP70, might be a useful intervention for patients with nonischemic heart failure to suppress cardiac remodeling and improve survival.

## 1. Introduction

Heart failure is a complicated, progressive clinical syndrome that results from impairment of either ventricular filling (diastole) or ejection of blood (systole). Patients with heart failure often experience symptoms of insufficient blood supply, such as fatigue and weakness, shortness of breath, and exercise impairment [1]. When cardiac function is <40%, as determined in an echocardiographic evaluation, patients are diagnosed with systolic heart failure, which is also known as heart failure with reduced ejection fraction (HFrEF) [1,2]. It is widely accepted that the main cause of systolic heart failure or HFrEF is a loss of myocardial tissue due to ischemic heart disease or myocardial infarction [1,2]. However, a notable portion of HFrEF is caused by nonischemic diseases, such as inflammatory cardiomyopathy, hereditary hypertrophic cardiomyopathy, and long-lasting arrhythmia [1,2]. As the heart supplies oxygen and nutrition to the periphery by pumping blood, failing hearts tend to compensate for inadequate hemodynamic demands by increasing chamber size and ventricular volume [3,4]. Although the remodeling of the heart chamber is initially beneficial, heart function gradually worsens, which in turn leads to permanent and irreversible deterioration remodeling known as dilated cardiomyopathy (DCMP) [5]. There are no available therapeutics for DCMP [6].

Heat shock proteins (HSPs) are a group of proteins that are activated in response to thermal shock and function to protect cells against damage [7,8]. Although most HSPs are constitutively expressed [7], certain subtypes are not detected under basal conditions but are robustly induced. HSPs are commonly induced by thermal stress, but they can also be activated by other stimuli [9,10,11,12]. HSPs were originally found to act as molecular chaperones to protect against protein denaturation under thermal shock. However, numerous studies have shown that HSPs can also function as scaffolding proteins for de novo synthesis [13], minimize DNA damage [14], and even regulate post-translational modifications [15].

Protein phosphatase 2A (PP2A) is a major phosphatase complex that is composed of three different subunits: the A subunit, which functions as the scaffold; the B subunit, which interacts with the substrate and thus determines substrate specificity; and the C subunit, which has catalytic activity [16,17]. As the C subunit functions as a phosphatase, PP2CA is sometimes referred to as PP2A. Cardiac-specific overexpression of PP2CA in CD1 [18] mice resulted in deterioration of cardiac remodeling, which led to increased chamber size and reduced ejection fraction (EF) [18,19]. Transgenic overexpression of PP2CA halved total cellular phosphorylation levels, and the proteins showing decreased phosphorylation included regulators of intracellular calcium handling [18,19].

In our previous study, we showed that PPP2R5A, the B subunit of PP2A, specifically interacts with HDAC2 and that transient overexpression of the catalytic subunit, PP2CA, reduced HDAC2 phosphorylation [20]. In addition, expression of HSP70 successfully neutralized the phosphatase activity of PP2CA by interfering with the interaction between PP2CA and HDAC2 [15]. Our study was terminated at a relatively young age, before 8 weeks of postnatal life, because it was already reported that chronic overexpression of PP2CA in the heart leads to global cardiac remodeling [18]. In the present study, we evaluated the role of HSP70 in a model of PP2CA-induced chronic heart failure by assessing cardiac function and survival in single (either HSP70 or PP2CA) and double transgenic mice for 6 months.

## 2. Materials and Methods

### 2.1. Reagents

Bovine serum albumin (BSA), 2,2,2-tribromoethanol, paraformaldehyde, and dithiothreitol were purchased from Sigma (St. Louis, MO, USA). Phosphatase inhibitor and protease inhibitor were obtained from Gendepot (Barker, TX, USA). Water was purified using a Milli Q system (Millipore, Molsheim, France). Ponceau S was obtained from Biosesang (Seongnam, Korea). Masson’s trichrome staining kit was purchased from Abcam (Cambridge, UK).

### 2.2. Antibodies

The antibodies used were as follows: anti-phospho-serine, anti-methyl-lysine, anti-acetyl-lysine, anti-phospho-Akt1 (Ser473), anti-Akt1, anti-phospho-troponin I (cardiac) (Ser23/24), anti-troponin I, anti-phospho-STAT5 (Tyr694), anti-STAT5, anti-phospho-phospholamban (Ser16/Thr17), anti-phospholamban, anti-phospho-p38 (Thr180/Tyr182), anti-p38, anti-phospho-STAT3 (Tyr705), anti-STAT3, anti-phospho-ERK1/2 (Thr202/Tyr204), and anti-ERK1/2 (all from Cell Signaling Technology, Danvers, MA, USA); anti-PP2CA (Abcam, Cambridge, UK); anti-HSP70 (Enzo Biochem, New York, NY, USA); and anti-HSP90 and anti-GAPDH (Santa Cruz Biotechnology, Santa Cruz, CA, USA).

### 2.3. Animal Model

The animal experiments were approved by the Chonnam National University Medical School Research Institutional Animal Care and Use Committee (CNU IACUC-H-2017-31). Cardiac function of the transgenic mice was assessed by ultrasound at 6 months, after which serial measurements of cardiac function were terminated, according to the animal usage protocol. Survival was assessed three times a week, and the date of death was recorded. Mice with natural death were excluded from the experiments by applying an exclusion criterion to avoid possible bias due to cold ischemic time variance. Mice data were strictly processed according to animal experiment guidelines and none of them were randomly excluded.

### 2.4. Genetically Engineered Mice

The two transgenic mouse lines expressing rat HSP70 and PP2CA were described previously [18,20,21]. The HSP70 line expresses rat inducible wild-type HSP70 (Hspa1a: heat shock 70 kDa protein 1A) under the control of the chicken beta-actin promoter and a cytomegalovirus enhancer, which allows overexpression of HSP70 in the heart and skeletal muscle in the C57BL/6 background [21]. In the PP2CA transgenic mice, the whole coding sequence from mouse PP2CA was expressed from the cardiac-specific promoter myosin heavy chain 6 with the SV40 transcriptional terminator in the CD1 background [18]. To avoid the confounding factor of different backgrounds, data were collected from each of the F1 offspring mice, which were obtained from the mating of the two transgenic mouse lines. Transgenic overexpression was constant, regardless of the fur color of F1 offspring.

The genotype of each mouse was confirmed by PCR using the following specific primers:

TgHSP70

sense: 5′-CTGGACTCTAACACGCTGGCTGAGAAAG-3′;

antisense: 5′-CTGAACTCCGGAGAGAAGGAGTAAC-3′.

TgPP2CA

sense: 5′-CCCTTACCCCACATAGACC-3′;

antisense: 5′-CTTAAACACTCGTCGTAGAACC-3′.

### 2.5. Cross-Sectional Area Measurement

To assess the concentric remodeling of the myocardium in vivo, the cross-sectional area of the cardiac muscle fibers was measured. Each heart was fixed with 3.7% paraformaldehyde, and Masson’s trichrome staining was performed. Horizontally sectioned myotubes in the left ventricular free wall were used for the calculation. The circumference of the muscle fiber was measured using automated software (NIS-Elements AT program; Nikon, Tokyo, Japan). If the longest diameter was more than 2-fold greater than the shortest diameter, the fibers were excluded because it was considered to be a longitudinal or oblique section.

### 2.6. Echocardiography

Cardiac function was measured by ultrasonography (Vivid S5; General Electric, Chicago, IL, USA). Mice were anesthetized with 2,2,2-tribromoethanol (300 mg/kg, intraperitoneal injection), which was checked by the lack of a response to a light touch. At the papillary muscle level, 2-dimension M-mode was acquired from the parasternal long-axis view or parasternal short-axis view. EF was determined using the Teichholz formula: EF(%) = (Vd − Vs)/Vd, where Vd is the LV volume at the end of diastole and Vs is the LV volume at the end of systole, and Vd = [7/(2.4 + LVIDd)] × LVIDd^3^, Vs = [7/(2.4 + LVIDs)] × LVIDs^3^, where LVIDd is the LV interventricular dimension at the end of diastole and LVIDs is the LV interventricular dimension at the end of systole.

### 2.7. Quantitative Real-Time Polymerase Chain Reaction

Total mRNA was extracted with TRIzol (Thermo Fisher Scientific, Waltham, MA, USA). cDNA was synthesized using random hexamers (M-MLV reverse transcriptase, Thermo Fisher Scientific, Waltham, MA, USA). Quantitative real-time PCR was carried out using the QuantiTect SYBR Green kit (Qiagen, Hilden, Germany) with a Rotor-Gene Q (Qiagen, Hilden, Germany). PCR analysis was performed in triplicate, and the average was regarded as a single result. Relative mRNA transcript contents were normalized to GAPDH. The specific oligomer sets used were as follows:

Mouse GAPDH

sense: 5′-GCATGGCCTTCCGTGTTCCT-3′;

antisense: 5′-CCCTGTTGCTGTAGCCGTAT-3′.

Mouse α-MHC

sense: 5′-GCTGGAAGATGAGTGCTCAGAG-3′;

antisense: 5′-CCAGCCATCTCCTCTGTTAGGT-3′.

Mouse β-MHC

sense: 5′-GCGGACAAAGGCAAAGGCAAGGCAAA-3′;

antisense: 5′-ATGCAGCGTACAAAGTGAGGGTGCGT-3′.

Mouse ANP

sense: 5′-CTGGGCTTCTTCCTCGTCTTGGC-3′;

antisense: 5′-CCTGCTTCCTCAGTCTGCTCACTCA-3′.

Mouse BNP

sense: 5′-TTATCTGTCACCGCTGGGAGGTC-3′;

antisense: 5′-GAGGGTGCTGCCTTGAGACCG-3′.

Mouse ACTA2

sense: 5′-ACTCTCTTCCAGCCATCTTTCA-3′;

antisense: 5′-ATAGGTGGTTTCGTGGATGC-3′.

Mouse COL1A1

sense: 5′-CTCCTGACGCATGGCCAAGA-3′;

antisense: 5′-TGGGTCCCTCGACTCCTACA-3′.

Mouse SERCA2

sense: 5′-GTGAAGTGCCATCAGTATGACGG-3′;

antisense: 5′-GTGAGAGCAGTCTCGGTAGCTT-3′.

Mouse PP2CA

sense: 5′-CGAGGGAATCACGAGAGCAG-3′;

antisense: 5′-GTGCTCGGATGTGATCCAGT-3′.

Mouse HSP70

sense: 5′-CCGTGGAGGACGAGGGTCTCAAG-3′;

antisense: 5′-GCCCCGAAGCCCCCAGCC-3′.

### 2.8. Statistics

Statistical significance was analyzed using PASW Statistics 26 (SPSS, IBM, Chicago, IL, USA). One-way analysis of variance (ANOVA) with LSD post hoc testing was used to compare the four groups. Survival rate curves were visualized using Prism 9.2 (GraphPad, San Diego, CA, USA) with the Kaplan–Meier method. The Mantel–Cox test was applied to measure significance. Significance was determined at *p* < 0.05.

## 3. Results

### 3.1. Expression of HSP70 Improves Cardiac Dysfunction and Survival Rates of TgPP2CA Mice

Gergs et al. reported that chronic expression of PP2CA in TgPP2CA mice slightly increased total heart weight and reduced cardiac contractility, with dilated ventricles [18]. Using our previous experimental model, we observed that HSP70 blocked PP2CA-induced negative regulation in the development of cardiac hypertrophy [20]. Based on our observations, we wondered whether specific expression of HSP70 in the heart could alleviate PP2CA-induced chronic heart failure.

We measured cardiac function until postnatal week 20–21 using echocardiography and calculated survival rates (Figure 1). The cardiac function of PP2CA-expressing transgenic mice (TgPP2CA) started to decrease from postnatal week 7, and gradual impairment continued until termination of follow-up. Interestingly, the double transgenic mice, which express both HSP70 and PP2CA, showed significantly retarded deterioration of EF when compared to TgPP2CA mice (Figure 1A). Both EF and fractional shortening were dramatically reduced in TgPP2CA mice, whereas cardiac function was less reduced in the dTg mice (Figure 1B,C). In 90% of the TgPP2CA mice, the EF was less than 40%, a generally accepted threshold for a failing heart. In contrast, the proportion of failing hearts was diminished to 50% in the dTg mice (Figure 1D). The TgPP2CA mice abruptly died at around 4 months of postnatal life due to systemic congestion caused by a failing heart, and expression of HSP70 improved survival rates (Figure 1E).

### 3.2. HSP70 Attenuates Ventricle Enlargement Caused by PP2CA

We further evaluated the geometry parameters of the transgenic mice at 24 weeks. Left ventricular internal diameter, which was measured at the papillary muscle level, was dramatically increased in TgPP2CA mice both at diastole (LVIDd) and systole (LVIDs). It is interesting that both the LVIDd and LVIDs of the dTg were significantly ameliorated when compared to these values in TgPP2CA mice (Figure 2B,C), which was correlated with the finding shown in Figure 1, that contractile function was improved in dTg mice. LV volume was calculated in diastole and systole. Based on the changes in LVIDd and LVIDs, the LV cavity was robustly enlarged in TgPP2CA mice, suggesting chamber dilatation. This deterioration was notably ameliorated in dTg mice (Figure 2D,E), which implied that expression of HSP70 suppressed eccentric remodeling (Figure 2F) and partially prevented contractile dysfunction (Figure 1).

### 3.3. HSP70 Expression Improves the Dilated Cardiomyopathy Phenotype of TgPP2CA Mice

We observed that chronic expression of PP2CA in the heart in a mixed background led to LV dilatation and contractile dysfunction. We further evaluated whether HSP70 could neutralize the deterioration effect of PP2CA expression in dTg mice. The LV free wall thickness of TgPP2CA mice was significantly thinner than that in their nontransgenic control littermates. Interestingly, LV free wall thinning was attenuated in dTg mice (Figure 3A,B). Next, we euthanized the mice and measured their heart weights to confirm the hypertrophic phenotype. The heart weight/body weight ratio, a parameter of cardiac hypertrophy, was increased in TgPP2CA mice (third group, Figure 3C). We also observed myocardial enlargement in the cross-sectional area evaluation (third group, Figure 3D). Strikingly, HSP70 successfully reduced these changes caused by PP2CA overexpression (fourth group, Figure 3C,D). Taken together, based on the results of cardiac function analyses (Figure 1) and geometry parameters (Figure 2 and Figure 3), we concluded that chronic overexpression of PP2CA accelerates DCMP, whereas coexpression of HSP70 ameliorates fatal remodeling and prevents cardiac dysfunction.

### 3.4. Cardiac Gene Expression

Next, we performed quantitative real-time PCR to assess any potential changes in the expression of marker genes related to heart failure or fibrosis. As expected, gene expression related to hypertrophic adaptation (α-myosin heavy chain (Figure 4A) and β-myosin heavy chain (Figure 4B)), ventricular wall stress (atrial natriuretic peptide (Figure 4C) and B-type natriuretic peptide (Figure 4D)), and cardiac fibrosis (smooth muscle alpha-actin (Acta2, Figure 4E) and type 1 collagen (Col1a1, Figure 4F)) was dramatically increased in TgPP2CA mice. However, HSP70 failed to repress the increased expression of these genes.

### 3.5. HSP70 Participates in Post-Translational Modification

We showed that HSP70 improves the nonischemic DCMP phenotype caused by chronic overexpression of PP2CA. Next, we aimed to delineate the molecular mechanism by which HSP70 attenuates LV remodeling. HSP70 is a well-known chaperone that regulates protein folding and maintains protein homeostasis. We also demonstrated that HSP70 specifically bound to phosphorylated proteins and maintained substrate phosphorylation by interfering with their recognition by phosphatases. Hence, we hypothesized that HSP70 could participate in protein post-translational modifications. Therefore, we briefly checked whether HSP70 modulates protein modifications. We assessed the global changes in phosphorylation (Figure 5A), methylation (Figure 5B), and acetylation (Figure 5C) by Western blotting and confirmed that HSP70 regulated phosphorylation and acetylation.

Since PP2CA functions as a phosphatase, we focused on the changes in phosphorylation. Several proteins have been reported as substrates of PP2CA, including troponin I and phospholamban. Dephosphorylation of these proteins is related to the modifications that occur in the transition to heart failure. We first evaluated whether HSP70 could modulate dephosphorylation. Analyses using phospho-specific antibodies revealed that HSP70 did not reverse the changes in phosphorylation caused by PP2CA expression. PP2CA reduced phosphorylation of cardiac troponin I (cTnI) and phospholamban (PLN) even in the presence of HSP70 (Figure 6A,B). Hence, we next sought to assess the changes in the phosphorylation of two known HSP70 substrates. PI3K/AKT and STAT3/5 are well-established substrates of activated HSP70, and the reduced phosphorylation levels of AKT (Figure 6C) and STAT5 (Appendix A) induced by PP2CA expression were significantly recovered in dTg mice (Figure 6). HSP90, a cochaperone molecule of activated HSP70, was expressed regardless of the presence of HSP70 (Figure 6D). HSP90 was increased in single or double transgenic mice. We also questioned whether HSP70 restored the activity of sarcoplasmic reticulum calcium ATPase (SERCA2a). Total amount of SERCA2a was significantly lowered in the presence of PP2CA as reported [18]. We observed that HSP70 partially improved the expression level of SERCA2a (Figure 6F).

To clarify the causality of favorable outcome in the development of failing heart due to phosphorylation of AKT, phosphorylation of AKT was further tested in young adult hearts whose cardiac function was preserved in the normal range. Similar to 6-month-old mouse hearts, phosphorylation of AKT was well retained in 7-week-old mice (Appendix A).

## 4. Discussion

In the present work, we revealed an HSP70-mediated salvage pathway in failing hearts driven by chronic overexpression of PP2CA in the heart. Previously, PP2CA was reported to exacerbate cardiac hypertrophy and contractile dysfunction through impairment of calcium handling [18,19]. Here, we demonstrated that long-lasting overexpression of HSP70 successfully ameliorated systolic dysfunction, LV global remodeling, and sudden cardiac death in TgPP2CA mice.

Chronic expression of PP2CA decreased the phosphorylation levels of PLN and cTnI, two cardiac-specific proteins related to intracellular Ca^2+^ regulation and contraction [18]. PLN is a muscle-specific SERCA inhibitor. Phosphorylation of PLN attenuates its SERCA inhibitory function, which activates SERCA, resulting in a lower cytoplasmic Ca^2+^ concentration [22]. In patients with heart failure, reduction in *p*-PLN levels was observed [23]. cTnI is a cardiac-specific protein that regulates myocardial contraction and relaxation cycles. Phosphorylation of cTnI at Ser23/24 induces Ca^2+^ desensitization of myofilaments to steady state and causes relaxation. It was reported that cTnI-Ser23/24 phosphorylation is decreased in patients with dilated heart failure [24]. Decreased p-PLN levels lead to a high Ca^2+^ concentration in the cytosol, and decreased p-cTnI levels cause myofilament contraction. Taken together, these results indicate that the chronic PP2CA stimulus disrupts Ca^2+^ homeostasis and subsequent cardiac contraction, which leads to systolic failure.

Our results showed that TgPP2CA mice had a cardiac hypertrophy phenotype (Figure 3) with an enlarged chamber cavity (Figure 2). Moreover, the LV free wall thickness was significantly lower than that in their nontransgenic littermates (Figure 3). Based on these findings, we can conclude that chronic overexpression of PP2CA induces DCMP. It is noteworthy that overexpression of HSP70 in PP2CA-expressing mice successfully alleviated the deterioration to DCMP. HSP70 improved systolic heart failure, survival, LV global remodeling, and LV free wall thinning. The predominant beneficial effect of HSP70 expression in the heart was improved systolic function. It was previously reported that HSP70 knockout mice showed contractility impairment, suggesting that HSP70 is important for contractility [25]. However, the role of HSP70 in the heart is somewhat controversial. Transgenic expression of HSP70 alone did not induce cardiac hypertrophy, but HSP70 did accelerate cardiac dilatation and remodeling in a model of muscle-restricted coiled-coil (MURC) hypertrophy with RhoA-induced heart failure and atrial fibrillation [21] and provided no protection in a model of DCMP induced by mammalian sterile-like kinase 1 and reduced phosphoinositide 3-kinase [26]. Conversely, mice in which HSP70 was knocked out or chemically inhibited were resistant to β-adrenergic-induced cardiac hypertrophy [15,27]. Moreover, blocking HSP70 with a neutralizing antibody significantly ameliorated doxorubicin-induced left ventricular dilation and systolic dysfunction [28]. Contrary to these previous reports, we showed a protective mechanism of HSP70 against dilated cardiomyopathy, which requires further study.

It is interesting that cardiac marker gene expression in dTg mice was not correlated with the outcomes of dTg mice in terms of the DCMP phenotype. We did not observe any significant difference between the TgPP2CA and dTg mice, which may be related to the selection bias of included subjects. A portion of the TgPP2CA mice (23%) were excluded from the study due to severe systolic dysfunction. Although we included the echocardiography results from all TgPP2CA mice, those with severe systolic dysfunction were excluded from the real-time PCR and Western blot analyses. As the cold ischemic time in dead TgPP2CA mice was not identical to that in euthanized mice, we only utilized heart samples acquired from euthanized mice. In other words, relatively fewer affected TgPP2CA mice were utilized for the mRNA and protein analyses. Because we failed to demonstrate significant differences in mRNA expression, it is worth considering this variable.

As PP2CA primarily functions as a serine/threonine phosphatase [17], we focused on the differences in protein phosphorylation in the transgenic mice. We previously reported that HSP70 regulates phospho-HDAC2, which is a target of PP2CA [15]. HSP70 preferentially bound to phospho-HDAC2 and preserved its phosphorylation, which was induced by hypertrophic stimuli, such as β-adrenergic signaling and pressure overload. In this study, we tested several candidate genes reported in previous studies. As systolic heart failure with chamber dilatation in TgPP2CA mice was significantly ameliorated in dTg mice, we first evaluated essential proteins related to intracellular calcium handling. Contrary to our expectation, HSP70 did not restore the phosphorylation levels of these proteins (Figure 6). Instead, the phosphorylation levels of AKT was largely preserved by transgenic expression of HSP70. This selective function of HSP70 suggested that it did not directly regulate or interfere with the phosphatase activity of PP2CA. Of note, we already stated that there is no evidence of a physical interaction between HSP70 and PP2CA [20].

AKT plays a critical role in cardiac growth and physiological function. Furthermore, it is generally accepted that AKT preserves cardiac function and protects the heart against devastating diseases. For example, AKT-deficient mice show exacerbated cardiac hypertrophy due to pressure overload [29]. Phosphorylation of AKT has also been closely linked to diabetes mellitus; reduced phosphorylation of AKT was observed in patients with diabetes [30], and the disease and its complications were resolved when its phosphorylation was restored [31,32]. In our study, decreased levels of phospho-AKT were observed and were correlated with systolic failure and DCMP, and phospho-AKT levels were largely preserved in dTg mice, along with preserved systolic function and attenuated global remodeling. Although the specific signal transduction pathway downstream of AKT is not yet known, AKT could be an important target for restoration of DCMP.

Overexpression of HSP70 alone improved systolic dysfunction and DCMP in TgPP2CA mice. As we did not confirm activation of a signal transduction pathway following expression of HSP70 in the dTg mice, unbiased approaches to clarify the role of HSP70 in this phenotype are needed.

## 5. Conclusions

Chronic activation of HSP70 in the heart may be a useful target for intervention in nonischemic heart failure.

## Figures and Tables

**Figure 1 cells-10-03180-f001:**
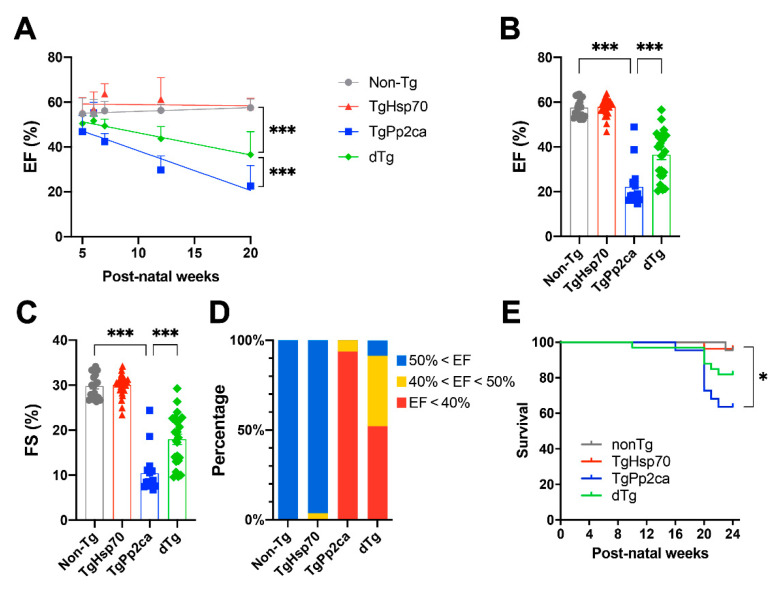
HSP70 alleviated PP2CA-induced cardiac dysfunction and subsequent death. (**A**) The serial changes in the ejection fraction (EF) of transgenic mice were measured. Note that the abrupt decrease in EF was significantly improved in double transgenic mice expressing both HSP70 and PP2CA (dTg). (**B**,**C**) Cardiac function of individual mice was measured at 20 weeks postnatal using echocardiography. Both EF (**B**) and fractional shortening (FS) were dramatically reduced in TgPP2CA mice (blue squares), whereas these decreases in contractile function were retarded in dTg mice (green rotated squares). (**D**) Heart failure caused by PP2CA was improved in HSP70-expressing mice. The EF of many TgPP2CA mice was below 40%, which indicates systolic heart failure. In contrast, the proportion of mice with heart failure was markedly reduced among dTg mice. (**A**–**D**) Every dot indicates a single mouse. Cohort size, nTg: 18, TgHSP70: 27, TgPP2CA: 16, dTg: 23. (**E**) Kaplan–Meier survival curves of transgenic mice. During the 6-month cohort study, sudden death of the TgPP2CA mice was observed after 15 weeks, which was correlated with the deterioration of cardiac function, as shown in (**A**). Survival was significantly better in dTg mice than in TgPP2CA mice by Mantel–Cox rank test. Cohort size, nTg: 22, TgHSP70: 27, TgPP2CA: 22, dTg: 33. One-way analysis of variance (ANOVA) with LSD post hoc testing was applied to confirm the significance of differences in panels (**A**–**C**). A single asterisk * indicates *p* < 0.05; *** indicates *p* < 0.001.

**Figure 2 cells-10-03180-f002:**
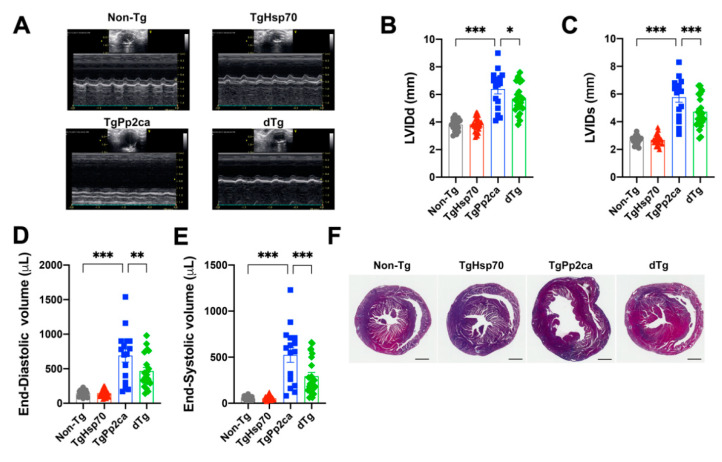
Cardiac geometry in echocardiogram measurements. (**A**) Representative M-mode echocardiograms from a mouse in each group. (**B**,**C**) Ventricular enlargement. Left ventricular (LV) internal diameter at diastole (LVIDd, **B**) and systole (LVIDs, **C**), which represents the LV chamber size at the papillary muscle level. LV cavity enlargement was predominant in TgPP2CA mice but was attenuated in dTg mice. (**D**,**E**) Global remodeling. End-diastolic volume (**D**) and end-systolic volume (**E**) in each group of transgenic mice. Fatal LV remodeling observed in TgPP2CA mice was ameliorated in dTg mice. (**B**–**E**) Cohort size, nTg: 18, TgHSP70: 27, TgPP2CA: 16, dTg: 23. (**F**) Representative heart morphology. Scale bars = 1 mm. Statistical significance was assessed using ANOVA with LSD post hoc test. Asterisks indicate statistical significance: * *p* < 0.05; ** *p* < 0.01; *** *p* < 0.001.

**Figure 3 cells-10-03180-f003:**
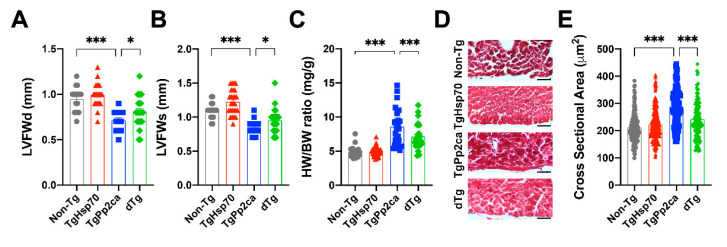
Overexpression of HSP70 prevents PP2CA-driven dilated cardiomyopathy. (**A**,**B**) Echocardiograms revealed that the LV free wall of TgPP2CA mice was significantly thinner than that of their non-Tg littermates, and this change was reversed in dTg mice. Cohort size, nTg: 18, TgHSP70: 27, TgPP2CA: 16, dTg: 23. (**C**) Overall increase in heart weight was observed in TgPP2CA mice, without any changes in body weight, while it was attenuated in dTg mice. Cohort size, nTg: 33, TgHSP70: 42, TgPP2CA: 27, dTg: 25. (**D**) Representative images of the LV free wall. Scale bars = 5 μm. (**E**) Measurement of the cross-sectional area. Mild cardiac hypertrophy was observed in TgPP2CA mice, and HSP70 suppressed this change. Cell counts were assessed from 5 individual mice in each group, nTg: 145, TgHSP70: 197, TgPP2CA: 93, dTg: 110. Data were analyzed using ANOVA with LSD post hoc testing. Asterisks indicate statistical significance: * *p* < 0.05; *** *p* < 0.001.

**Figure 4 cells-10-03180-f004:**
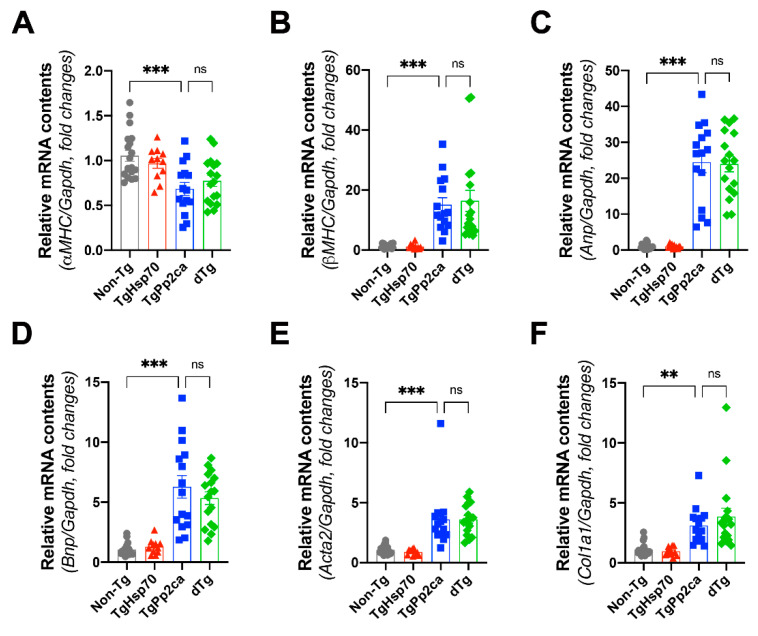
Relative changes in cardiac marker gene expression. (**A**,**B**) Fetal gene reprogramming: alpha myosin heavy chain (**A**) and beta myosin heavy chain (**B**) are indicators of the hypertrophic response. (**C**,**D**) Natriuretic peptides: atrial (**C**) and B-type (**D**) natriuretic peptides are indicators of LV wall stress. (**E**,**F**) Fibrosis markers: smooth muscle alpha-actin (**E**) and type 1 collagen (**F**). HSP70 failed to reverse the changes in marker gene expression. Cohort size, nTg: 19, TgHSP70: 11, TgPP2CA: 15, dTg: 17. ANOVA and LSD post hoc testing were performed. Asterisks indicate statistical significance: **, *p* < 0.01; ***, *p* < 0.001; ns, not significant.

**Figure 5 cells-10-03180-f005:**
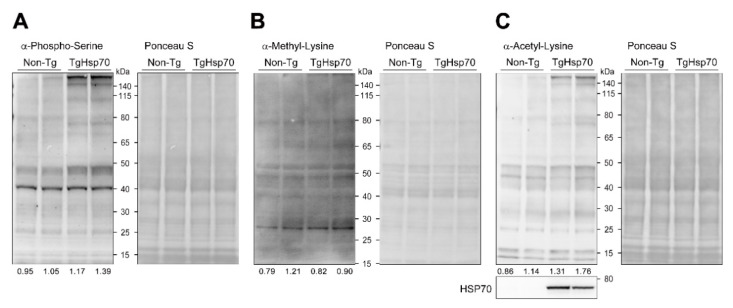
HSP70 modulates post-translational modifications. (**A**–**C**) Protein modifications in heart lysates obtained from TgHSP70 mice were compared to those in lysates from non-Tg littermates; phospho-serine (pSer, **A**), methyl-lysine (MeK, **B**), and acetyl-lysine (AcK, **C**). Both serine phosphorylation and lysine acetylation were significantly increased in TgHSP70 hearts. Phosphorylation was predominantly changed. The numbers at the bottom of each figure indicate the results of densitometer analysis.

**Figure 6 cells-10-03180-f006:**
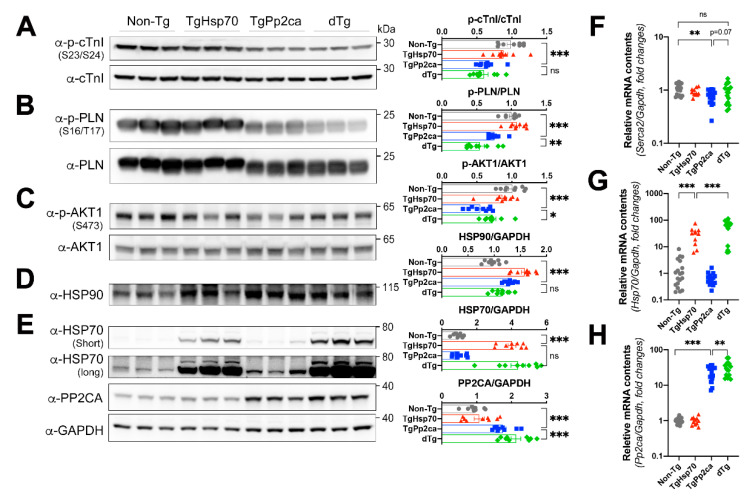
Phosphorylation levels of select proteins in hearts from each group of transgenic mice at 6 months of age. (**A**,**B**) Calcium handling factors. Known PP2CA substrates related to the failing heart phenotype were assessed. Phosphorylation of cardiac troponin I (Ser23/24) (**A**) and phospholamban (Ser16/Thr17) (**B**) was significantly reduced in TgPP2CA mice, and expression of HSP70 did not affect these changes in phosphorylation. Known factors associated with HSP70 were also assessed, and the phosphorylation levels of protein kinase B (Ser473) (AKT1, **C**). (**D**) HSP90, an important binding partner of HSP70, was measured. HSP90 levels were significantly increased in transgenic mice. Western blotting (**E**) and real-time PCR (**G**,**H**) also confirmed transgenic overexpression of PP2CA and HSP70 in the relevant groups. (**F**) Individual dot plots beside Western blot depict densitometer analysis. Transgenic expression of PP2CA attenuated SERCA2 mRNA levels in the heart, whereas transgenic HSP70 expression partially restored it. (**F**–**H**) Cohort size, nTg: 19, TgHSP70: 11, TgPP2CA: 15, dTg: 17. One-way ANOVA and LSD post hoc testing were performed. * *p* < 0.05; ** *p* < 0.01; *** *p* < 0.001; ns, not significant. Displayed figures are representative images of three independent experiments.

## Data Availability

Not applicable.

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
