# Peer review of "Overexpression of Heat Shock Protein 70 Improves Cardiac Remodeling and Survival in Protein Phosphatase 2A-Expressing Transgenic Mice with Chronic Heart Failure"

_cells, 2021, doi:10.3390/cells10113180_

Round 1

Reviewer 1 Report

1) The authors should properly quantify each  western blot presented in Figure 6A-I by performing statistical analyses. Without statistical analyses the data is meaningless. The appropriate quantification is important to gain insight about the mechanisms involved in Hsp70 cardioprotection.

Author Response

We thank Reviewer 1 for the constructive comments on our manuscript.

1. Quantification of Western blot images

1) The authors should properly quantify each western blot presented in Figure 6A-I by performing statistical analyses. Without statistical analyses the data is meaningless. The appropriate quantification is important to gain insight about the mechanisms involved in Hsp70 cardioprotection.

Response: We understand the reviewer’s points. We already performed three independent experiment sets and presented representative images in main figure. We have collected every single densitometer result and applied statistical analyses. Now the quantification of all the densitometry data is available in supplemental figure 1.

Reviewer 2 Report

Even though the authors have added some more details in the methods section, this study still doesn't fully address the major consents brought up in the initial review. In addition, the overexpression of Hsp70 only showed a very modest impact on cardiac function, and the authors used a very high number of samples to show statistical significance. Those are critical weaknesses that prevent the conclusion drawn in the study.

Author Response

We thank Reviewer 2 for the constructive comments on our manuscript.

  1. Modest effect with huge sample size

Even though the authors have added some more details in the methods section, this study still doesn't fully address the major consents brought up in the initial review. In addition, the overexpression of Hsp70 only showed a very modest impact on cardiac function, and the authors used a very high number of samples to show statistical significance. Those are critical weaknesses that prevent the conclusion drawn in the study.

Response: We understand reviewer’s points. Our findings from double transgenic mice expressing Pp2ca and Hsp70 demonstrated that Hsp70 modestly improved contractile dysfunction which was induced chronic overexpression of Pp2ca. However, we observed that sudden cardiac death from Pp2ca transgenic group by deterioration of global remodeling. As we described in the main text, we excluded those mice from cohort because of possible bias from cold ischemic time variance. In other words, several mice whose cardiac function was severely low were excluded from the main data and the mice were more prominent in TgPp2ca group. We clarify our exclusion criteria we have applied in the method section.

We also understand reviewer’s concern regarding sample size. Our animal usages were approved by the Chonnam National University Medical School Research Institutional Animal Care and Use Committee (CNU IACUC-H-2017-31). According to guideline for animal experiment, we could not “select” or “drop” animal data acquired during cohort study except natural death. We cordially ask reviewer’s understanding regarding our limitation. 

Round 2

Reviewer 1 Report

My recommendation is to accept the manuscript

Author Response

Thank you.

Reviewer 2 Report

Even though the authors provided some explanation, this paper has fundamental issues related to the experimental design and data interpretation. The changes in some groups are modest and may not have any biological meanness.

Author Response

We understand limitaion of our study. We have revised several data according to editor's comments. We hope current version satisfies reviewer.

This manuscript is a resubmission of an earlier submission. The following is a list of the peer review reports and author responses from that submission.

Round 1

Reviewer 1 Report

The study aimed to determine the role of Hsp70 on heart failure. The authors created an Hsp70 and PP2CA overexpression transgenic mice model. They measured the survival rate, cardiac function, pathological changes of heart failure, and the expression of cardiac biomarkers. They found that overexpression of Hsp70 partially reversed PP2CA-induced heart failure. This study demonstrated some evidence on the association of Hsp70 and heart failure.

However, there are some major issues.

  1. Hsp70 is a chaperone protein which universally expressed at a high level. It's unclear why there was no detectable level of Hsp70 in Non-Tg and dTg mice (Fig. 6H). This study also lacks controls to determine the impact of basal Hsp70 on heart failure. Because of this, the data from this study is hard to provide solid evidence to support the conclusions drawn by the authors.
  2. Lack of details about the process of developing transgenic mice.
  3. Provide the sample size for all figures.

Reviewer 2 Report

In this work, Yoon and colleagues investigated the impact of Hsp70 overexpression in a model of non-ischemic heart failure using protein phosphatase 2 catalytic subunit  A expressing transgenic (TgPp2ca) mice. Through a series of in vivo studies and molecular data the authors conclude that chronic activation of Hsp70 improved the dilated cardiomyopathy phenotype of TgPp2ca mice, suggesting that Hsp70 may be a useful target for intervention in non-ischemic heart failure. The topic is interesting and novel. However, a number of concerns arose during the review of this article; these concerns are listed below in order of appearance in the manuscript.

1) Abstract line 21: The following sentence is vague: "No notable expression changes were noted by real-time quantitative PCR." The sentence should be removed or rewritten to include more detailed information.

2) Methods; section 2.2: The authors don´t cite the specific phosphorylation sites that were evaluated in each protein. This information is crucial for interpreting the data, and should be included in the manuscript.

3) Figure 1, panel E: The number of mice used to generate the Kaplan Meier curve should be reported.

4) Hsp90 is an important binding partner of Hsp70. How are the levels of Hsp90 in Hsp70 overexpressing mice?

5) The authors should include in the manuscript the quantification of the western-blot data presented in Figure 5. This information is crucial for interpreting the data.

6) Information regarding the phosphorylation sites evaluated should be included in the figure. For example, PLN can be phosphorylated at two different sites, Ser16 or Thr17, which one was evaluated? The same applies to AKT, and the others.

7) The authors should include in the manuscript the quantification of the western-blot data presented in Figure 6. It is very difficult to interpret the molecular data, especially without quantification.

8) Given the echo data showing the improved function and attenuation of hypertrophic phenotype it would be important to evaluate SERCA levels in addition to phosphorylated-PLN.  This result would help to understand the beneficial mechanisms induced by Hsp70 overexpression in the TgPp2ca mice.